# Deep Learning for Microstructural Characterization of Synchrotron Radiation-Based Collagen Bundle Imaging in Peri-Implant Soft Tissues

Nicole Riberti [1] , Michele Furlani [2] , Emira D'Amico [3] , Luca Comuzzi [4] , Adriano Piattelli [5,6] , Giovanna Iezzi [3] and Alessandra Giuliani [2,*]

1 Neurosciences Imaging and Clinical Sciences Department, University of Chieti-Pescara, 66100 Chieti, Italy
2 Odontostomatologic and Specialized Clinical Sciences Department, Università Politecnica delle Marche, Via Brecce Bianche 12, 60131 Ancona, Italy
3 Medical, Oral and Biotechnological Sciences Department, University of Chieti-Pescara, 66100 Chieti, Italy
4 Private Practice, via Raffaello 36/a, 31020 San Vendemiano, Italy
5 School of Dentistry, Saint Camillus International University for Health Sciences (Unicamillus), 00131 Rome, Italy
6 Facultad de Medicina, UCAM Universidad Catolica San Antonio de Murcia, Av. de los Jerónimos, 135, Guadalupe de Maciascoque, 30107 Murcia, Spain
* Correspondence: a.giuliani@univpm.it; Tel.: +39-071-220-4603

**Abstract:** The study of the organizational kinetics in the area surrounding the transmucosal part of dental implants promises to ensure an accurate diagnosis of the healing process, in terms of osseointegration and long-term implant success. In this demonstrative work, the morphological, qualitative and quantitative characteristics of 3D images of collagen bundles obtained by synchrotron-based high-resolution X-ray tomography were analyzed. Data analysis was performed using deep learning algorithms, neural networks that were applied on multiple volumes extracted from connective portions of different patients. The neural network was trained with mutually consistent examples from different patients; in particular, we used a neural network model, U-Net, well established when applying deep learning to datasets of images. It was trained not only to distinguish the collagen fibers from the background, but also to subdivide the collagen bundles based on the orientation of the fibers. In fact, differently from conventional thresholding methods, deep learning semantic segmentation assigns a label to each pixel, not only relying on grey level distribution but also on the image morphometric (shape or direction) characteristics. With the exception of Pt2 biopsies that, as confirmed by the polarized light investigation, were shown to present an immature tissue condition, the quantity, the anisotropy degree and the connectivity density of transverse bundles were always demonstrated to be higher than for longitudinal ones. These are interesting and new data; indeed, as collagen bundles are organized in an intertwining pattern, these morphometric and 3D complexity parameters, distinguished in transversal and longitudinal directions, give precise indications on the amount and distribution of connective tissue forces exerted during the healing process.

**Keywords:** wound; collagen; dental implant; imaging; artificial intelligence; deep learning

## 1. Introduction

Wound healing is an essential process for restoring tissue integrity after trauma. While large skin wounds often heal with hypertrophic scarring, resulting in disfigurement and reduced joint mobility, these adverse healing outcomes were not generally observed in oral mucosa, which generally heals faster than skin [1]. Several studies have identified differences between oral and cutaneous wound healing [2–5]. It has been observed that the faster wound closure in the oral mucosa compared to that in skin could be due to the presence of saliva, a faster immune response, and increased extracellular matrix remodeling.

In particular, during remodeling, oral wounds show less contraction and better restoration of tissue architecture in terms of collagen structure [1].

However, in comparison to skin, much less is known about the kinetics of remodeling during oral mucosa healing [6]. In particular, understanding the organizational kinetics of the area surrounding the transgingival part of dental implants would ensure more accurate monitoring of the healing process, with favorable effects in terms of implant osseointegration and long-term success.

Until now, tissue tension during wound healing has been attributed to cellular forces produced by tissue-resident (myo-)fibroblasts alone; more recently, the storage of tensile forces in the extracellular matrix has been shown to have a significant, hitherto overlooked, contribution to macroscopic tissue tension. In particular, it has been shown that the organization of collagen fibrils is related to tissue contraction, indicating a mechanical contribution of the collagen fibrils to tension in the contraction process [7,8].

In this direction, reorganization of the soft tissues in the early stages following the placement of the dental implant was recently imaged using high-resolution tomography based on synchrotron radiation, discovering how the collagen bundles of the newly formed connective tissue intertwine impart tensile strength to the entire peri-implant soft tissue. It was, therefore, hypothesized that good organization of collagen bundles may reduce the risk of early bone resorption by reducing the infiltration of inflammatory cells [9]. However, the limitation of the previous synchrotron study was related to the fact that conventional techniques of segmentation of the tomographic image are based on the discrimination of grey levels, i.e., the physical density of the tissues. Therefore, conventional segmentation techniques fail to discriminate the orientation of the collagen fibers, as they all have the same physical density.

In recent years, artificial intelligence (AI) has been applied to different types of images (MRI [10], tomography [11,12], X-ray and ultrasound [13]) and for different purposes (detail detection, object recognition and noise reduction [14]). This allowed the various disciplines to interface with new methodologies and to analyze images and data with a support they had not previously considered [15]. In particular, AI has also established itself as a support tool in dentistry [16], especially in facilitating diagnosis and treatment planning [17]. Recently, AI has been used to improve image interpretation in 2D and 3D dental radiology [18], identifying dental caries [19] and automatically segmenting anatomical structures such as maxilla, mandible and teeth [20,21].

In this demonstrative study, for the first time to the authors' knowledge, AI was applied directly on highly resolved images acquired by synchrotron tomography to automatically segment collagen bundles of the peri-implant connective tissue. We trained the algorithm starting from available data, the so-called training data, and then we made it acquire the ability to predict new information, called test data. Once the basic model was set up, the artificial neural networks were able to distinguish the internal portions of the soft tissue not only according to the grey levels of the synchrotron image, as conventional thresholding methods do, but also according to the orientation of the collagen bundles themselves. In this way, we were able to quantitatively distinguish longitudinal and transversal peri-implant collagen bundles, with evidence of the times and modalities of connective tissue formation around the implant during the wound healing process.

## 2. Materials and Methods

### 2.1. Samples

The peri-implant soft tissues of four patients (posterior jaw) were retrieved from the archives of the Dental School of the University of Chieti-Pescara, Italy.

They were selected based on the specific characteristics listed in Table 1.

**Table 1.** Description of the investigated samples. *Ctr1_st_wt*: immediately loaded and retrieved after 6 weeks (*st:* short time; *wt:* immediately loaded). This specimen had already been studied [9,12]. *Ctr2_lt_no-wt:* the abutment was inserted subgingivally, healed without loading for 6 months and then placed on the surface with the healing screw; the final prosthesis was made after another 3 months, with a total of 9 months before applying the masticatory load (*lt:* long time; *no-wt:* not immediately loaded). *Pt1_st_no-wt:* patient test 1—not loaded and retrieved after 6 weeks. *Pt2_st_no-wt_r:* patient test 2—not loaded and retrieved after 8 weeks. Right side of the jaw (*r*); *Pt2_st_no-wt_l:* patient test 2—not loaded and retrieved after 8 weeks. Left side of the jaw (*l*).

| Sample Name | Time before Retrieval | Immediately Loaded | Region |
|---|---|---|---|
| *Ctr1_st_wt* | 6 weeks | YES | - |
| *Ctr2_lt_no-wt* | 10 years | NO | - |
| *Pt1_st_no-wt* | 6 weeks | NO | - |
| *Pt2_st_no-wt_r* | 8 weeks | NO | right |
| *Pt2_st_no-wt_l* | 8 weeks | NO | left |

The use of these specimens for these scientific purposes was approved by the Ethical Committee of the University of Chieti-Pescara (CODE: BONEISTO, 15 September 2019). Indeed, the implants included in this study had been archived in previous years.

Before being stored in the archives, the samples were fixed by immersion in 10% buffered formalin, dehydrated in an increasing series of alcoholic rinses and finally embedded in glycol-methacrylate resin (Technovit 7200 VLC; Kulzer, Wehrheim, Germany). After retrieval from the archives, they were processed according to Ref. [9]: briefly, they were sectioned along their longitudinal axis in order to obtain two portions: the first was examined by synchrotron-based microtomography after removal of the abutment, and subsequently sectioned along its longitudinal axis for histological analysis; the second portion was used to obtain histologic transversal sections of the peri-implant soft tissue.

*2.2. Synchrotron Radiation-Based Phase-Contrast Microtomography*

The microtomography experiment was performed at the SYRMEP beamline of the ELETTRA Synchrotron Facility (Basovizza, Italy); the following parameters were set: 1800 projections, each with 0.2 s exposure time, over a total range of 180°; peak energy at ~17 keV; sample-detector distance at 100 mm, achieving a pixel size of the projections of 890 nm. The high-resolution acquisition exploited the propagation-based phase-contrast setting, made possible by the coherence characteristics of the synchrotron beam. The refractive index n = $1 - \delta + i\beta$ was reconstructed, where the phase shift term $\delta$ is related to the electron density of the tissues inside the sample, and $\beta$ is proportional to the absorption index. The Paganin method [22] was used to retrieve the different phases (collagen, vessels, etc.), assuming a constant $\delta/\beta$ ratio of 100.

The next phase of image processing was the data elaboration: it was performed with the Dragonfly software (Vers. 2022.1; Object Research Systems, Montreal, QC, Canada) [23] and in particular with the deep learning tool, able to apply artificial intelligence to the segmentation process. Moreover, the open-source Fiji software [24] was used to perform a threshold image segmentation and the following extraction of quantitative morphometric parameters. In particular, the morphometric evaluation of the collagen bundles was calculated with the BoneJ plugin [25] of the Fiji software. It was performed by exploiting the structural indices usually measured in bone samples: collagen specific volume (expressed as a percentage), the degree of anisotropy and the connectivity density (expressed as pixel$^{-3}$). The percentage volume is the amount of collagen tissue per unit of volume considered; the degree of anisotropy gives information on the directionality of the collagen bundles. The more anisotropic an object is, the more the parameter will tend from zero to

one; the connectivity density quantifies how much collagen fibers are connected to each other, presenting an index of interlacing per unit of volume.

### 2.3. Training and Test Data

Four connective tissue volumes of $600 \times 300 \times 300$ pixels$^3$ (around 0.04 mm$^3$), taken at the interface with the implant, were selected from each sample to form a dataset as uniform as possible so as not to create a bias within it. The ORS Dragonfly software was used to manually segment slices with an artificial intelligence tool, which provides all of the instruments required to best separate fibers (longitudinal and transversal) and background and create a network that distinguishes three classes (Figure 1).

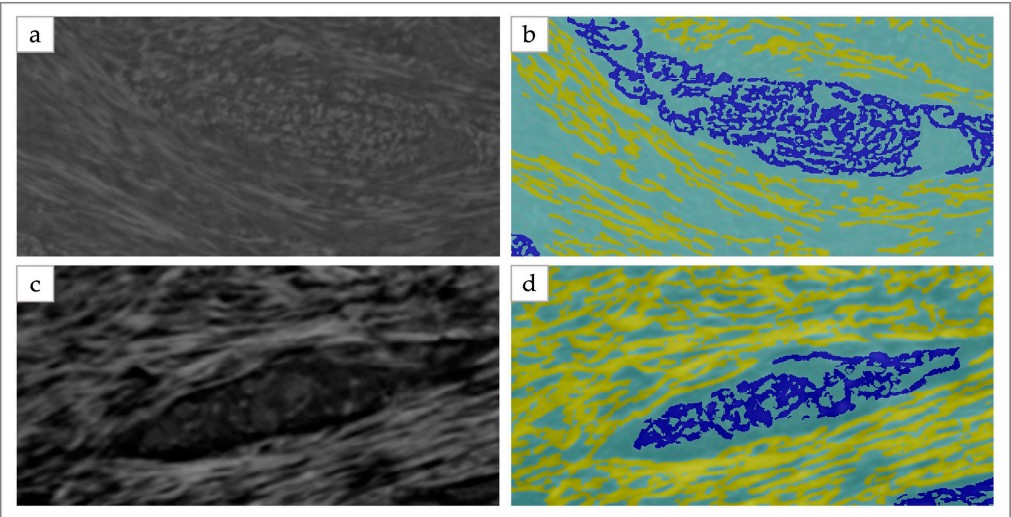

**Figure 1.** Manual segmentation in three classes (transversal fibers in yellow, longitudinal fibers in blue, and background in teal) with ORS Dragonfly AI tool. Ctr1_st_wt raw slice example (**a**), respective manual segmentation (**b**); Ctr2_lt_no-wt raw slice example (**c**), respective manual segmentation (**d**).

The training dataset was formed using a total of 50 slices as input to the algorithm: 10 slices were manually segmented in each of 3 volumes of Ctr1_st_wt and in each of 2 volumes of Ctr2_lt_no-wt. This training dataset was given as input to the AI tool, which internally separates training and validation. Training parameters chosen after several tests and set before starting the algorithm are reported in Table 2. The network was tested on all slices of each volume (including the ones remaining from those used for training) in order to visualize the network's performances when applied to a 3D volume.

The graphics card of the PC on which the training was conducted was an NVIDIA GeForce RTX 3070, Version 512.59.

Due to the scarcity of input datasets, the data augmentation method [26] was applied. In fact, it is usually applied when the input data are insufficient to train the network or when the dataset is not varied enough: with this method, we wanted to obtain a more robust and wider dataset, avoiding redundancies that could lead to overfitting. The method aims to "deform" the data by means of geometric techniques such as flipping, rotation, scaling, zooming, cropping, or techniques that deal with image enhancement such as gamma correction and histogram equalization. Thus, both the original and modified images are introduced as input into the algorithm. The neural network does not notice the difference and sees the two groups as unique and makes predictions on both. This method was used in the online version [27] inside the ORS Dragonfly AI tool to artificially increase the input dataset.

**Table 2.** Training parameters chosen after several tests and set before starting the algorithm.

| U-Net | |
|---|---|
| Class count | 3 |
| Depth level | 5 |
| Initial filter count | 64 |
| Patch size | 32 |
| Stride ratio | 1 |
| Batch size | 32 |
| Epochs number | 300 |
| Loss function | Categorical Cross-Entropy |
| Optimization algorithm | Adadelta |
| Metrics | Categorical Accuracy |
| Early stopping | NO |
| Reduce learning rate on plateau | YES |
| Training/Validation | 90/10 |

*2.4. Image Segmentation (by Thresholding as Comparative Method)*

Two segmentation algorithms, aiming to separate different image features based on different properties, were used: the Otsu histogram-based threshold technique [28] and the deep learning semantic segmentation method.

The former is based on the choice of a segmentation threshold applied to the grey level histogram that separates pixels into two groups that have more similar grey levels (proportional to the physical densities—mg/cm$^3$), usually foreground and background [29]. Otsu calculates the maximum variance between them and sets the threshold on this value [30]. In this study, the Otsu algorithm was applied to each volume selected (consisting of 300 slices), and it chose a single threshold for each of them.

Conversely, the deep learning semantic segmentation method assigns a label to each pixel, not only relying on grey level distribution but also on the image morphometric characteristics. Thus, if objects inside the image have a different shape or direction, they will be classified as two distinct subgroups, and, therefore, associated with a different color. The use of the semantic segmentation method involves neural networks; we chose to use U-Net. It was developed by Olaf Ronneberger et al. [31] and was designed for biomedical image segmentation. The architecture is divided into two parts; the first is called encoder, which is the contracting path to capture context in the image and find key elements, while the second is the symmetric expanding path called decoder, which employs the learned pooling layers to enable precise localization of features.

*2.5. U-Net Structure and Parameters*

U-Net corresponds to the network predominantly used in biomedical applications; the reference model is the one proposed by Olaf Ronneberger et al. [31].

After several tests, we chose the training parameters listed in Table 2. The patch size is a subsection of the image on which the convolution kernel iterates, while the stride ratio indicates how much that kernel moves through the different patches of the image. The batch size defines the number of slices that will be propagated through the network at each training step. Categorical Cross-Entropy is the loss function chosen to understand the performance of the algorithm as a function to be minimized [32]. The optimization algorithm is Adadelta, and the metric is Categorical Accuracy. To ensure the robustness of the model and prevent overfitting, 10% of input slices were used as validation; we selected a small portion because of the limited number of images. The epochs number was fixed to 300 to guarantee the uniqueness of all evidence; consequently, Early Stopping

was deactivated. Data Augmentation was chosen to secure enough input slices and to differentiate them; in that way, geometric applications such as flipping, rotation, shearing and scaling are made to the images and added to the standard dataset by convincing the network that they are new incoming images.

### 2.6. Comparative Light Microscopy Investigations

Longitudinal and transversal sections were obtained for histological evaluation. Histological examinations were performed using a light microscope (Laborlux S, Leitz, Wetzlar, Germany) connected to a high-resolution video camera (3CCD, JVCKY-F55B, JVC, Yokohama, Japan) and to a PC.

Birefringence was measured using polarized light microscopy to determine the orientation of the collagen bundles. An Axiolab optical microscope was used to examine thin transversal tissue sections (Laborlux S, Leitz, Wetzlar, Germany). Two linear polarizers and two quarter-wave plates were used in this instrument to transmit circularly polarized light. Because of the variation in the existing light refraction, collagen bundles aligned transversely to the light propagation direction, i.e., parallel to the section plane, appeared bright, whereas collagen fibers aligned along the light propagation axis, i.e., perpendicular to the plane of the section, appeared in a different color because refraction did not occur.

### 2.7. Statistical Methods

The following tests were used for statistical analysis: Welch *t*-test (deep learning vs. thresholding segmentation and transversal vs. longitudinal collagen bundles), Brown–Forsythe and Welch ANOVA tests and Dunnett's multiple comparisons between samples for each morphometric parameter (only deep learning data). These tests were preferred to the classic ANOVA in order to also include data failing equivariance conditions. The tests were executed by GraphPad Prism 6.0 (GraphPad, Inc., San Diego, CA, USA).

## 3. Results

The aim of the semantic segmentation technique was to label each image pixel with a corresponding color that indicates the membership class of what is being represented. Thus, it was possible to quantify the collagen bundles and reconstruct the pattern of the connective tissue during the wound healing in the peri-implant site. This information is of paramount relevance in order to understand the action of forces in this site and, eventually, how the presence of the masticatory load can cause the collagen tissue to adapt to better respond to these stresses.

### 3.1. U-Net: Accuracy and Loss for Training and Validation

The U-Net network was trained and validated by following its performance epoch by epoch and verifying that the training was moving in the right direction, i.e., that the loss function decreases (Figure 2a) and the accuracy function (Figure 2b) increases until an acceptable level is reached. In this way, it was proved that the network learned without incurring overfitting [33].

At the end of the process (300 epochs), an accuracy of 83% was achieved on training and 87% on validation. The latter was higher most probably because of the absence of regularization techniques (that were applied in the training phase); the reason behind this choice was to avoid overfitting by making the network more generic at the expense of a lower accuracy level.

The accuracy already reached the regime in the first 100 epochs, exceeding 80%: this percentage can be considered a good level of learning, taking into account the scarcity of the samples [34].

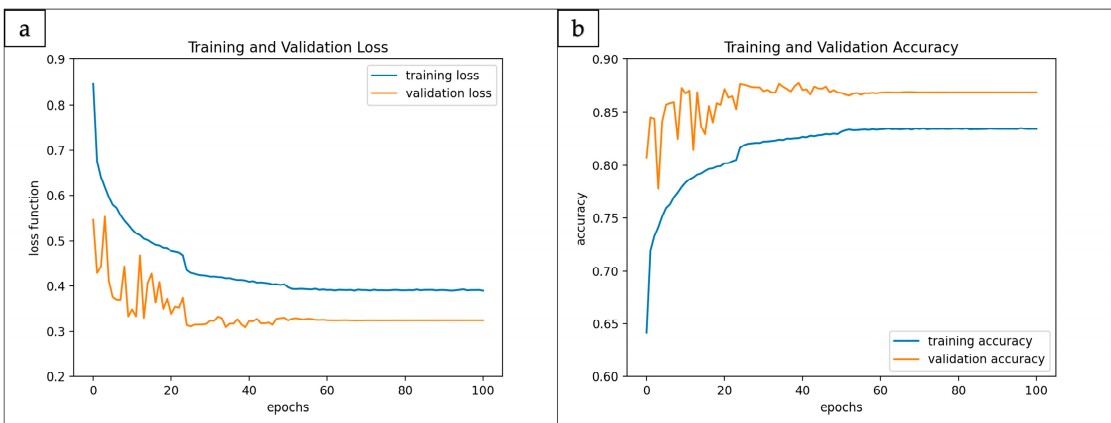

**Figure 2.** (**a**) The loss function minimization; (**b**) the accuracy trend; training and validation cases were plotted. Only the first 100 epochs were plotted because the network achieved its best performance around this level.

### 3.2. Semantic vs. Otsu Thresholding Segmentation

Semantic and Otsu thresholding segmentation were compared in order to verify if the results obtained by AI with the semantic segmentation method were superimposable with those obtained by common thresholding techniques.

The thresholding segmentation was applied using the open-source Fiji software [24]. As the first step, the Histogram Equalization Fiji plugin was used to enhance contrast in the spatial domain, adjusting the pixel intensity from 0 to 255 [35,36]. Subsequently, the Otsu segmentation process was chosen as the thresholding segmentation method (Figure 3a,c,e).

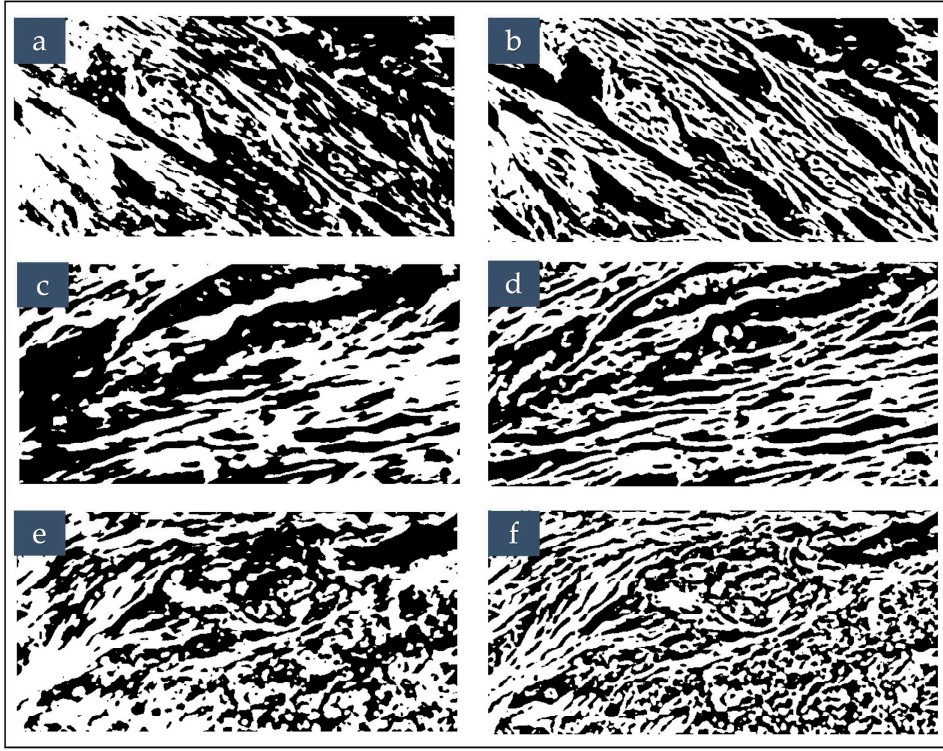

**Figure 3.** Thresholding (**a**,**c**,**e**) vs. semantic (**b**,**d**,**f**) segmentation methods. Examples of comparisons on homologous slices: (**a**,**b**) Ctr1_st_wt; (**c**,**d**) Ctr2_lt_no-wt; (**e**,**f**) Pt2_st_no-wt_l.

The semantic segmentation was performed with the Dragonfly software [23]. The semantic segmentation allowed us to distinguish not only background from connective tissue signals but also, unlike Otsu segmentation, transversal bundles from longitudinal bundles

(Figure 4 and Video S1 in Supplementary Materials); this was achieved with a training based on three classes. However, to compare the semantic with the Otsu thresholding method, in the former, the two classes of collagen bundles (transversal and longitudinal) were grouped and just distinguished from the background (Figure 3b,d,f). The following morphometric parameters were considered: the percentage volume, the anisotropy degree, and the connectivity density. They were represented, in Python, by means ± standard deviation (Table 3) of the four volumes extracted from each sample (Figure 5).

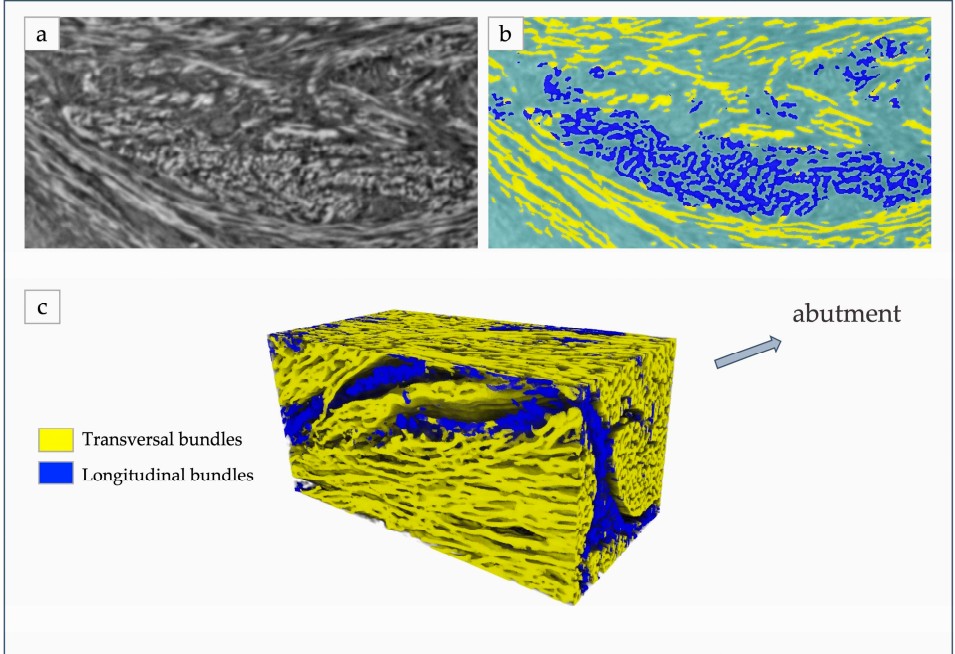

**Figure 4.** (**a**) Raw slice of a Ctr1_st_wt subvolume, as achieved after phase-retrieval reconstruction; (**b**) semantic segmentation of the same slice with deep learning: transversal bundles (yellow), longitudinal bundles (blue) and background (teal) were detected. (**c**) Application of artificial intelligence to a Ctr2_lt_no-wt subvolume and automatic segmentation of collagen bundles (blue: longitudinal; yellow: transversal).

**Table 3.** Welch *t*-test between deep learning vs. thresholding segmentation; Brown–Forsythe and Welch ANOVA test, Dunnett's multiple comparisons between samples per parameter (only deep learning data). Test executed by GraphPad Prism 6.0 (GraphPad, Inc., San Diego, CA, USA). c, •: $p < 0.05$; b, a, ••: $p < 0.01$; b,d, •••: $p < 0.001$.

| | Volume Percentage (%) | | Anisotropy Degree | | Connectivity Density ($\times 10^{-4}$ px$^{-3}$) | |
|---|---|---|---|---|---|---|
| | *Deep Learning* | *Thresholding* | *Deep Learning* | *Thresholding* | *Deep Learning* | *Thresholding* |
| **Ctr1_st_wt** | 40.3 ± 8.8 | 47.9 ± 1.8 | 0.691 ± 0.049 | 0.627 ± 0.040 | **3.48 ± 0.33** [a] | 2.34 ± 0.85 |
| **Ctr2_lt_no-wt** | **53.8 ± 1.9** • | **47.5 ± 3.0** • | 0.753 ± 0.040 | 0.771 ± 0.056 | 2.83 ± 0.22 •••,b | **1.08 ± 0.36** ••• |
| **Pt1_st_no-wt** | 50.7 ± 1.7 | 47.4 ± 2.4 | 0.818 ± 0.014 | 0.809 ± 0.029 | **4.00 ± 0.47** ••,c | 1.57 ± 0.05 •• |
| **Pt2_st_no-wt_r** | 52.3 ± 9.5 | 50.1 ± 2.6 | 0.649 ± 0.066 | 0.657 ± 0.050 | **3.45 ± 0.91** ••,d | **1.00 ± 0.24** •• |
| **Pt2_st_no-wt_l** | 51.8 ± 1.1 | 49.5 ± 2.6 | 0.430 ± 0.143 | 0.428 ± 0.121 | 7.75 ± 0.80 •••,a,b,c,d | **2.53 ± 0.34** ••• |

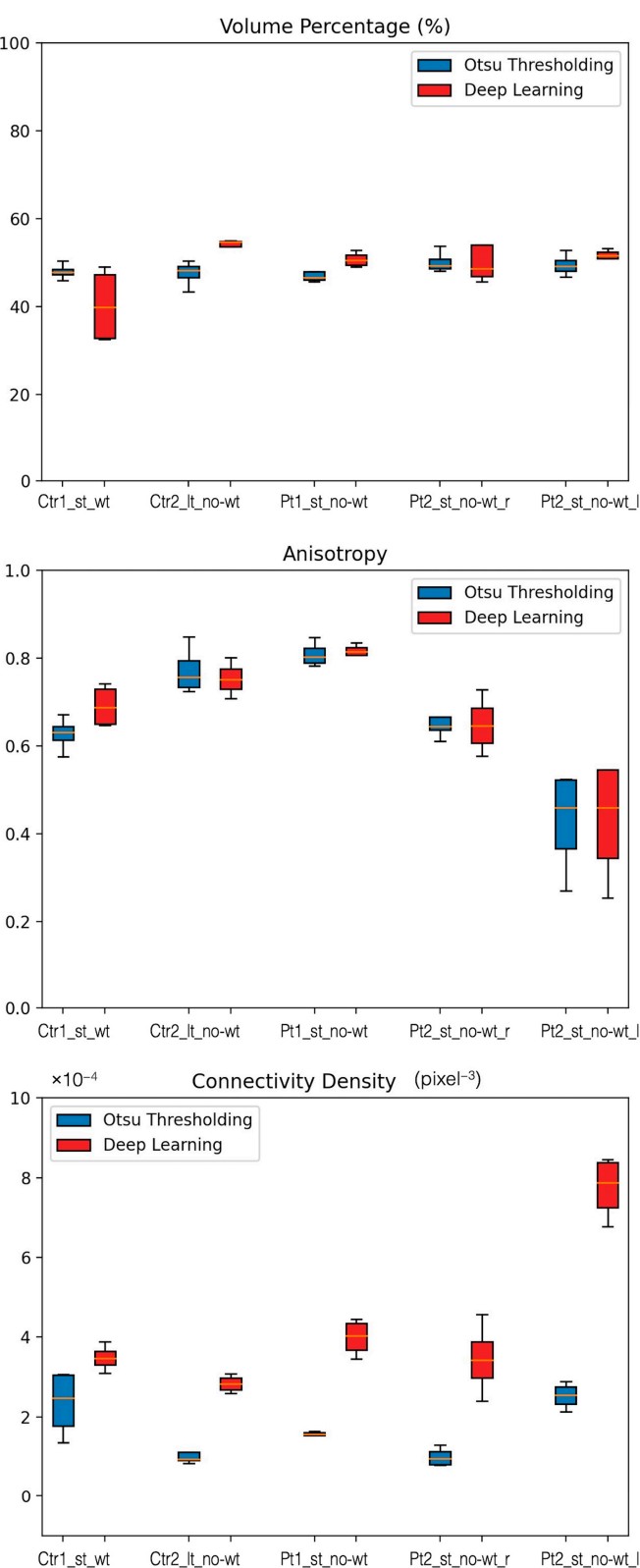

**Figure 5.** Box plots of volume percentage, degree of anisotropy and connectivity density calculated on each volume segmented with deep learning semantic method (red) and Otsu thresholding (blue). For deep learning, both longitudinal and transversal fibers were considered together for comparison.

Referring to the volume percentage, in general, the average values obtained by deep learning through semantic segmentation were similar to those obtained by the thresholding segmentation method. Interestingly, the volume percentage distribution throughout the

whole sample of Ctr2_lt_no-wt was significantly wider using semantic segmentation than in the case of the Otsu thresholding method.

The degree of anisotropy trend was found to be similar using the thresholding or the semantic segmentation techniques.

Conversely, referring to the connectivity density, the average values obtained by deep learning through semantic segmentation were significantly higher than those obtained by the Otsu thresholding segmentation method in all of the samples except for the Ctr1_st_wt. This higher level of connectivity was probably due to a better discrimination of the single fibers; consequently, each intertwinement was better recognized.

Interestingly, using semantic segmentation, the connectivity density was significantly higher in Pt2_st_no-wt_l than in all of the other samples. This was most likely attributable to the immaturity of the connective tissue present in this biopsy, also demonstrated by the morphological evidence shown in Figure S1 of the Supplementary Materials. In fact, the abundant presence of micro vessels (Figure S1a) and the presence of collagen fibers still strongly disorganized (Figure S1b) at a three-dimensional level suggested the presence of a connective tissue only at the beginning of the remodeling phase, with a wound not yet completely healed at the connective level.

### 3.3. Transversal vs. Longitudinal Bundles

Since semantic segmentation allows one to distinguish transversal bundles from longitudinal ones, the quantitative analysis of these two classes of bundles was performed on samples segmented by deep learning. Thus, the transversal and longitudinal collagen bundles, isolated by the neural network, were quantitatively analyzed in four different subvolumes extracted from each sample. This information could not have been obtained by thresholding segmentation, which merely analyzes the image based on grey levels (i.e., on the physical density mismatches) without looking at the morphology of the tissue itself. This quantitative analysis, made possible by semantic segmentation that discriminates transversal from longitudinal bundles, was reported by mean $\pm$ standard deviation (Table 4) and by box plots (Figure 6).

**Table 4.** Welch *t*-test between morphometric data in transversal vs. longitudinal collagen bundles; Brown–Forsythe and Welch ANOVA test, Dunnett's multiple comparisons between samples per parameter (only deep learning data). Test executed by GraphPad Prism 6.0 (GraphPad, Inc., San Diego, CA, USA). a,b,c,d,•: $p < 0.05$; ••: $p < 0.01$; •••: $p < 0.001$; ••••: $p < 0.0001$.

| Samples | Volume Percentage (%) | | Anisotropy Degree | | Connectivity Density ($\times 10^{-4}$ px$^{-3}$) | |
|---|---|---|---|---|---|---|
| | *Transversal* | *Longitudinal* | *Transversal* | *Longitudinal* | *Transversal* | *Longitudinal* |
| Ctr1_st_wt | **28.2 $\pm$ 8.9** • | **11.9$\pm$ 6.3** • | **0.747 $\pm$ 0.014** • | **0.619 $\pm$ 0.065** • | **2.32 $\pm$ 0.70** • | **0.90 $\pm$ 0.60** • |
| Ctr2_lt_no-wt | **48.0 $\pm$ 3.6** •••• | **5.8$\pm$ 2.2** a,•••• | **0.787 $\pm$ 0.037** • | **0.609 $\pm$ 0.108** • | **2.10 $\pm$ 0.49** •• | **0.30 $\pm$ 0.31** d,•• |
| Pt1_st_no-wt | **45.4 $\pm$ 2.5** •••• | **5.3$\pm$ 1.6** •••• | **0.834 $\pm$ 0.017** ••• | **0.747 $\pm$ 0.009** b,••• | **2.77 $\pm$ 0.31** ••• | **0.12 $\pm$ 0.11** ••• |
| Pt2_st_no-wt_r | **38.6 $\pm$ 13.0** • | **13.8 $\pm$ 10.6** • | 0.720 $\pm$ 0.065 | **0.719 $\pm$ 0.058** c | 1.87 $\pm$ 0.98 | 1.14 $\pm$ 1.34 |
| Pt2_st_no-wt_l | 30.2 $\pm$ 6.7 | **21.6$\pm$ 5.7** a | **0.647 $\pm$ 0.036** • | **0.497 $\pm$ 0.065** b,c,• | 1.96 $\pm$ 0.58 | **4.24 $\pm$ 1.52** d |

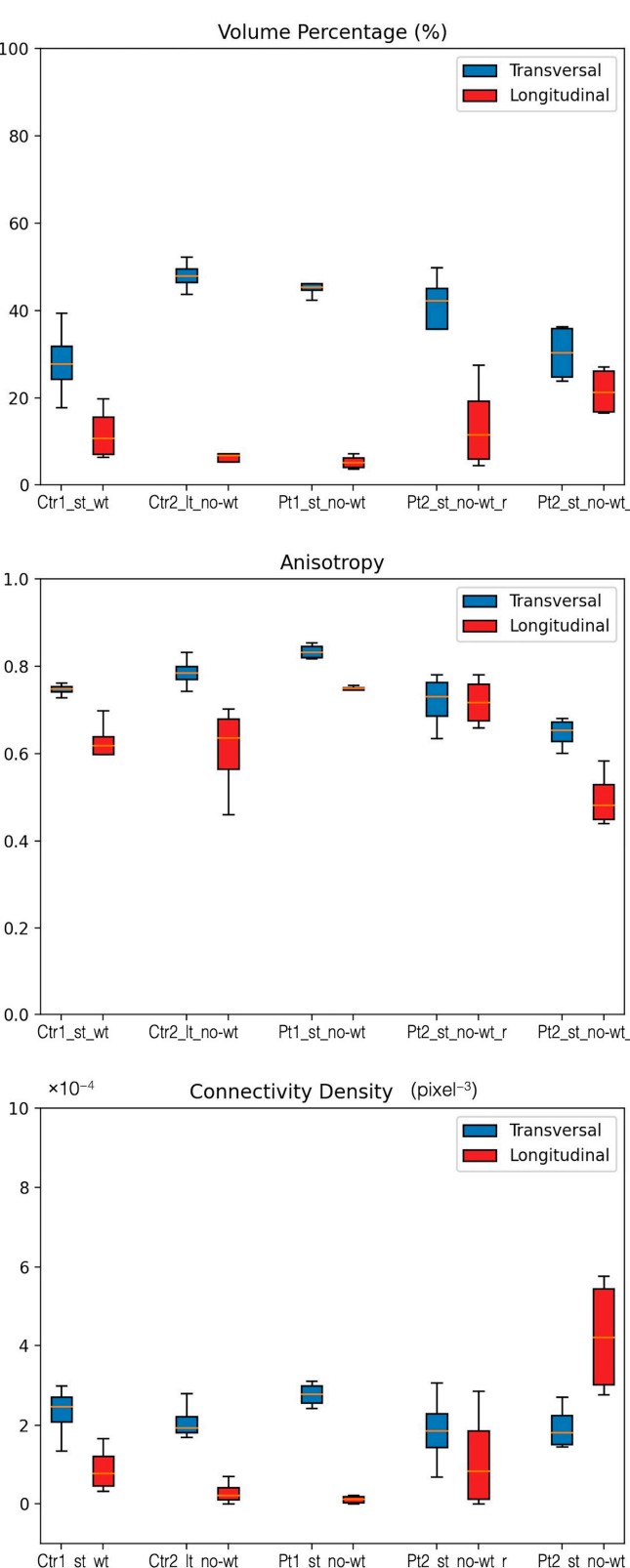

**Figure 6.** Box plots of volume percentage, degree of anisotropy and connectivity density calculated on each volume segmented with deep learning: transversal fibers (blue), longitudinal fibers (red).

In terms of volume percentage, the quantity of transverse bundles was always shown to be higher than for longitudinal ones.

The anisotropy degree was found to be significantly lower for longitudinal bundles than for transversal ones in all of the samples but Pt2_st_no-wt_r. Interestingly, Pt2_st_no-wt_l presented the lowest degree of anisotropy for both longitudinal and transversal bundles.

With regard to the connectivity density parameter, almost all samples, with the unique exception of those referred to patient 2, showed significantly higher values in the transversal direction than in the longitudinal one. Notably, the connectivity density of longitudinal bundles in the Pt2_st_no-wt_l sample was found to be considerably larger than in the other samples.

In general, referring to Pt2 (i.e., both Pt2_st_no-wt_r and Pt2_st_no-wt_l samples), high standard deviations were found, both considering the volume percentage and the connectivity density; this was not the case with Ctr1, Ctr2 and Pt1 samples. As previously observed, the biopsies extracted from patient 2, in particular the one referring to the left side of the jaw, had very particular characteristics, with high volume percentages and connectivity in the longitudinal direction and low anisotropy degree. This morphometric evidence was compatible with a probable immaturity of the connective tissue (already highlighted in Figure S1 of the Supplementary Material and probably linked to the patient's clinical history).

Polarized light microscopy confirmed the semantic segmentation results referred to the detection of transversal and longitudinal collagen bundles. It showed that the peri-implant mucosa was constituted by different collagen bundles, distinguishable both in transversal (Figure 7a–e) and longitudinal (Figure 7f–j) sections.

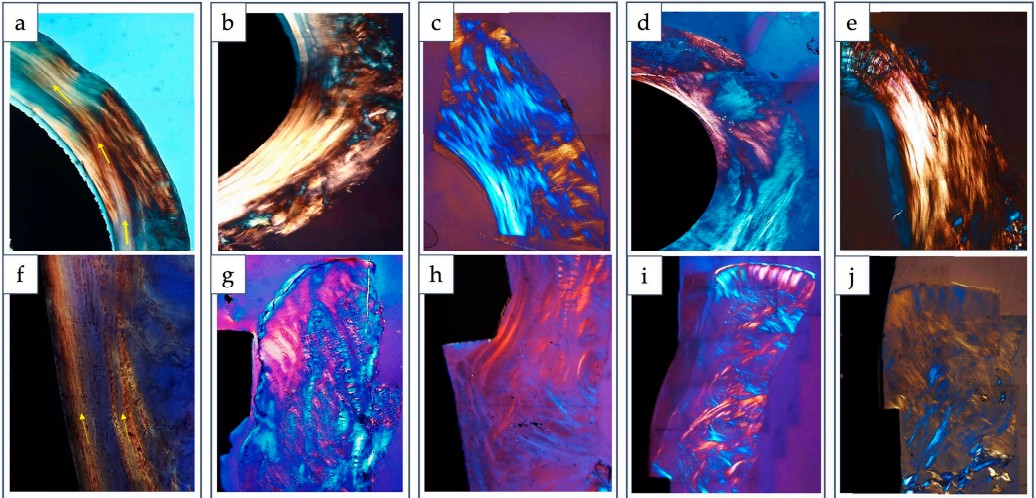

**Figure 7.** Polarized light microscopy. Transversal sections (**a–e**): (**a**) Ctr1_st_wt showed bundles better organized; (**b**) Ctr2_lt_no-wt, (**c**) Pt1_st_no-wt, (**d**) Pt2_st_no-wt_r, (**e**) Pt2_st_no-wt_l showed a predominance of transversal bundles. Longitudinal sections (**f–j**): (**f**) Ctr1_st_wt and (**h**) Pt1_st_no-wt showed fibers parallel to the long axis of the implant abutment; a network composed by woven bundles was observed in (**g**) Ctr2_lt_no-wt; (**i**) Pt2_st_no-wt_r and (**j**) Pt2_st_no-wt_l (**a**,**f**) from Ref. [9], licensed under an open access Creative Commons CC BY 4.0 license.

Regarding transversal sections, it was possible to observe how the semi-circular fibers coming from different collagen bundles intersected with each other by showing the semi-circular orientation of the collagen fibers around the transversal abutment profile. In Ctr1_st_wt (Figure 7a), the bundles were better organized, and the transversal fibers were alternated with collagen fibers parallel to the long axis of the implant (dark areas), whereas in all other cases (Figure 7b–e), transversal bundles predominated.

Regarding longitudinal sections, only Ctr1_st_wt (Figure 7f) and partially Pt1_st_no-wt (Figure 7h) showed, near the surface, longitudinal bundles distributed parallel to the long axis of the implant abutment. In all other cases (Figure 7g,i,j), a network composed of woven bundles was observed.

## 4. Discussion

In this demonstrative study, artificial intelligence was used, for the first time to the authors' knowledge, to perform morphometric analysis on high-resolution synchrotron-based tomographic images of peri-implant connective tissue. The overall objective was to obtain reliable quantitative information on collagen bundles and their 3D distribution. We showed that deep learning semantic image segmentation can better identify the bundles compared to a common thresholding segmentation technique [37]. Notably, we had recently found, by synchrotron imaging, the presence of transversely and longitudinally oriented (with respect to the dental implant axis) collagen bundles in the peri-implant soft tissue; these grow and organize intertwining patterns during their development at different times and in different quantitative distributions depending on environmental conditions, to date not fully understood [9]. Therefore, discriminating and quantifying transversal and longitudinal bundles turns out to be an extremely interesting step forward in understanding the wound healing process in the peri-implant soft tissue and the role of collagen in it.

In this study, we demonstratively showed that a good way to find these features is to use the semantic segmentation method of synchrotron-based highly resolved tomographic images. With the support of neural networks and deep learning, it was possible to quantify structures in the samples that were not considered before. In particular, collagen bundles were identified by their orientation and not by their physical densities. This is fundamental in order to discriminate transversal and longitudinal bundles that, having the same physical density, could not have been discriminated using conventional thresholding segmentation techniques. In practice, we have managed to create a neural network able to separate longitudinal and transversal fibers by U-Net [30,31].

Within the limits of the extremely reduced sample size, we did not detect a statistically significant difference in volume percentage and degree of anisotropy between data obtained by conventional segmentation methods based on thresholding and the innovative ones based on artificial intelligence. On the other hand, as regards the connectivity density parameter, it was observed that the data obtained in deep learning were higher for all samples than the data obtained with conventional thresholding (statistically significant in all samples except for the Ctr1_st_wt); this fact is certainly attributable to a greater discrimination of the collagen bundles and their connectivity through artificial intelligence.

Based on quantitative comparison between transversal and longitudinal bundles, carried out in deep learning, we discovered an extremely interesting fact: considering all the parameters globally, a clear prevalence was observed of the quantity and connectivity of transversal bundles compared to longitudinal ones in all of the samples; moreover, transversal bundles were found to be more anisotropically oriented (i.e., they had a preferred orientation) than the longitudinal ones. These differences were, in most cases, statistically significant and were confirmed by polarized light microscopy observations.

This evidence could indicate that in the wound healing process in the peri-implant area, the transverse fibers are formed first, and only subsequently the longitudinal ones. In reality, having found in the Ctr2_lt_no-wt, investigated after permanence in vivo for 10 years, the same quantitative mismatch between transversal and longitudinal fibers as in the other samples, we could reasonably exclude that the lower presence of longitudinal fibers compared to transversal ones was due to the limited healing time. Thus, in general, we can state that, even in a mature and functional peri-implant connective tissue, the number of longitudinal fibers is always lower than the number of transversal ones; this information is completely new and has never been observed in the literature.

However, in the Ctr1_st_wt, Pt2_st_no-wt_r and Pt2_st_no-wt_l, there were, in terms of volume percentages, the greatest equilibria between transversal and longitudinal bundles. This could suggest a better stability and functionality of these samples, but, as confirmed by polarized light microscopy observations, referring to the other two morphometric parameters, i.e., the degree of anisotropy and the density of connectivity, it could be observed that both samples referring to patient 2, with particular reference to Pt2_st_no-wt_l, showed high variability both in terms of volume percentage and connectivity. Moreover,

Pt2_st_no-wt_l appeared to have bundles less oriented than the other samples and highly vascularized (Figure S1a), indicating that the tissue was still highly immature. Indeed, in three samples (Ctr2_lt_no-wt, Pt2_st_no-wt_r, Pt2_st_no-wt_l), the longitudinal sections obtained by polarized light microscopy showed collagen fiber orientation that was never perfectly parallel to the surface of the abutment.

In conclusion, within the limits of the sample size, this demonstrative study showed that the implant subjected to immediate loading, i.e., the Ctr1_st_wt, presented the most balanced distribution of collagen bundles (transversal vs. longitudinal). This evidence would suggest a more favorable stabilization of the soft tissues, because a better balancing between transversal and longitudinal intertwining bundles should be due to forces balancing each other in those directions. Consequently, immediate loading appears to be a good solution, in patients where it is possible [38], since it stabilizes the connective tissue around the neck of the implant, which thus becomes an effective barrier against the apical migration of inflammatory cells towards the bone, thus preventing implant failure. The positive effect of immediate loading was observed also in previous studies [39–41], but in all of these investigations the analysis was just focused on bone tissue without searching for correlations with the peri-implant connective tissue organization. Clearly, given the demonstrative nature of this study and the extremely small sample size, this new finding has to be verified within a wider sample size. Thus, in the next steps of this research, we will expand the dataset and use a 3D U-Net with more complex models applicable with larger amounts of data. This way, it will be possible to support this preliminary study and validate the artificial intelligence analysis method.

**Supplementary Materials:** The following supporting information can be downloaded at: https://www.mdpi.com/article/10.3390/app13074423/s1, Video S1: Semantic Segmentation: transversal vs. longitudinal collagen bundles; Figure S1: Synchrotron imaging evidence of the connective tissue immaturity in Pt2; (a) Pt2_st_no-wt_l transversal slice: collagen bundles (CT), bone residuals (BR) and a rich vascularity were detected; (b) Pt2_st_no-wt_l transversal slice (detail): the collagen bundles were isotropically oriented and often revealed a zig-zag pattern, typical of immature contexts; (c) Ctr2_lt_no-wt transversal slice (detail): the collagen bundles were anisotropically oriented with a smooth pattern, typical of mature contexts.

**Author Contributions:** Conceptualization, N.R. and A.G.; Data curation, N.R., M.F., E.D., L.C. and G.I.; Formal analysis, N.R., M.F. and A.G.; Funding acquisition, G.I. and A.G.; Investigation, N.R., M.F., E.D., G.I. and A.G.; Methodology, N.R., G.I. and A.G.; Project administration, A.G.; Resources, L.C., A.P. and G.I.; Software, N.R., M.F. and A.G.; Supervision, A.P. and A.G.; Validation, G.I.; Visualization, N.R.; Writing—original draft, N.R., E.D. and A.G.; Writing—review and editing, N.R., M.F., E.D., L.C., A.P., G.I. and A.G. All authors have read and agreed to the published version of the manuscript.

**Funding:** The synchrotron experiments (proposal id: 20200335) were funded by the Cooperation Program "Support to the Italian Users of ELETTRA". No other sources of study funding were used.

**Institutional Review Board Statement:** The use of these specimens for these scientific purposes has been approved by the Ethical Committee of the University of Chieti-Pescara (CODE: BONEISTO, 15 September 2019). Indeed, the implants included in this study had been archived in previous years.

**Informed Consent Statement:** Informed consent was obtained from all subjects involved in the study.

**Data Availability Statement:** Histological and morphometric data are contained within the article. The complete synchrotron imaging data presented in this study are available on request from the corresponding author; indeed, they are not publicly available due to their extremely large size.

**Acknowledgments:** We acknowledge Elettra Sincrotrone Trieste for providing access to its synchrotron radiation facilities and Giuliana Tromba for assistance in using beamline SYRMEP. The deep learning-based morphometric data, the 3D reconstructions and videos in Supplementary Materials for this paper were generated using Dragonfly software, Version 2022.1 for Windows, Object Research Systems (ORS) Inc, Montreal, Canada, 2020; software available at http://www.theobjects.com/dragonfly (accessed on 30 January 2023).

**Conflicts of Interest:** The authors declare no conflict of interest.

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
