# Peer review of "Deep Learning for Microstructural Characterization of Synchrotron Radiation-Based Collagen Bundle Imaging in Peri-Implant Soft Tissues"

_applsci, doi:10.3390/app13074423_

Round 1

Reviewer 1 Report

Review report for the review paper "Collagen bundles rearrangement during wound healing in peri-implant soft tissues: an innovative application of Artificial Intelligence to synchrotron radiation-based imaging"

General observation for the manuscript:

The article investigates using the Otsu histogram-based threshold technique and U-Net architecture as an AI model to characterize 3D images of collagen bundles for the first time, as the authors claim. Using statistical analysis, the authors compared the Otsu thresholding technique with the results from the U-Net model. They concluded that the U-Net provides better results for the segmentation tasks, expectedly. Also, the authors noticed the discrimination between the longitudinal and transversal bundles for the first time, which is very interesting to study further. Furthermore, they correctly used data augmentation to increase their limited dataset. However, the paper is not ready to be published as it is not organized well and has some serious flaws, especially in dataset preparation. The reviewer would strongly recommend that the authors carefully review the comments, implement them in their manuscript to increase the scientific sound of their article, and resubmit their article for the journal.

 Comments to authors:

The most important issue with the article is the dataset. The dataset is not elaborated, and the authors referred to another paper for dataset preparation. The dataset, how it is annotated, and how it was augmented need to be elaborated with graphic demonstration. Furthermore, what are the targets for the pixels in the volumes? Or have the authors used an unsupervised model to classify the pixels into 2 or sometimes 3 different classes? Because this has not been mentioned in the manuscript. If the implemented U-Net model is unsupervised, how can the accuracy and loss function be calculated? Please elaborate on whether this is an unsupervised U-Net model or how the pixels were annotated, and the targets were created. Furthermore, I suggest training other deep learning models to compare with the trained U-Net.

Also, please address the following comments as well:

1. In the abstract, please include how precisely the AI identified the collagen fibers. I suggest adding highlights of your research regarding numbers and comparing the results from AI to previous results in the abstract.

2. The introduction is too brief. Please elaborate more on the biomedical background of the research and why applying AI models to predict the wound healing process in the tissue around the implant is important.

3. Section 2.1: The dataset is not well-defined and difficult to follow. Please arrange the dataset more organized and explain or label them more appropriately, so the reader can follow the dataset more easily. I would suggest using a table to present the dataset as more organized.

Furthermore, in the samples section, it is required to elaborate on the segmentation classes. How many classes are there? Two classes? How were the targets created, and how would the AI model's performance be compared with the targets for each pixel?

4. Line 146: Is this threshold set for all tissue volumes, including the augmented data? Moreover, does this method set a threshold for each sample in the dataset individually, or evaluate samples together and then set the threshold? Please address these questions in the paragraph.

5. Line 151: Please consider reevaluating this paragraph, especially from lines 151-154.

6. Section 2.5: The structure, including the input size, number of layers, and depth of the layers of the U-Net, needs to be specified. I'd recommend adding a figure for the structure of the U-Net used in the article.

7. Authors correctly indicated that segmentation aims to label each image pixel with a corresponding color; however, they never showed the datasets and the targets for each pixel they are trying to reach. This is a serious flaw in their work as the readers expect a demonstration of the input dataset and the targets for each pixel. Please consider addressing this issue and elaborating on the dataset as well as the classes for them.

8. Line 275: why the connectivity density is significantly higher in Pt2_st_no-wt_l than in other samples?

9. Section 3.1: The fact that validation accuracy is higher than the training accuracy raises questions about the data selection for the input. Please elaborate on how the input data was randomized to avoid the model being biased. Also, normalizing the data could help the model perform better; maybe the authors could use normalized data as input and compare the results. Furthermore, why not use cross validation in evaluating the accuracy? This could help overcome the issue of having a small dataset.

In addition, using augmented data is not creating an entirely new dataset, as the images are still related. Therefore, it is very important to elaborate on the data augmentation types used. Moreover, was a sample from a patient kept separately from the training and validation dataset to be used as a test for unseen data? Or all the images from the dataset were used in training and validation?

10. Section 3.2: Do the results, especially in figure 4, indicate the Semantic segmentation performs better than the Otsu thresholding? Please care to elaborate on this section or the discussion.

11. Section 3.3: The purpose of the statistical analysis performed in this section is unclear to the reviewer. Are the transversal bundles and longitudinal bundles compared for each sub-volume? The results from figure 5 do not make too much sense. For example, do we have the background in volume percentage as well? If not, then the sum of the transversal and longitudinal bundles should be 100% which is not in this figure.

Using statistical analysis, are we trying to compare different properties of the sub-volume based on the transversal and longitudinal bundles identified by the AI segmentation?

12. Line 319: How did the polarized light microscopy confirm the semantic segmentation results? Maybe it would be better to add the results from the semantic segmentation alongside the microscopy images for comparison.

13. Line 403: The authors have drawn an interesting conclusion about the Ctr1_st_wt sample. However, this is very brief, and I would suggest discussing and emphasizing this conclusion with more details in their manuscript.

Author Response

Dear reviewer,

We thank you for the observations made to our study which allowed us to complete and improve it. We have answered all the questions point by point. We hope that the changes introduced will prove to be a valid answer to your questions.

  1. In the abstract, please include how precisely the AI identified the collagen fibers. I suggest adding highlights of your research regarding numbers and comparing the results from AI to previous results in the abstract. 

We included how precisely the AI identified the collagen fibers, with highlights on our research. We did not insert numbers because of the demonstrative nature of this study and the limited number of samples.

  1. The introduction is too brief. Please elaborate more on the biomedical background of the research and why applying AI models to predict the wound healing process in the tissue around the implant is important.

We expanded the biomedical background of the research, also introducing new recent references. We also explained the relevance of applying AI models to predict the wound healing process in the tissue around the implant.

  1. Section 2.1: The dataset is not well-defined and difficult to follow. Please arrange the dataset more organized and explain or label them more appropriately, so the reader can follow the dataset more easily. I would suggest using a table to present the dataset as more organized.

Furthermore, in the samples section, it is required to elaborate on the segmentation classes. How many classes are there? Two classes? How were the targets created, and how would the AI model's performance be compared with the targets for each pixel?

In Section 2.1 a Table (new Table 1) was added to present the dataset, as from your suggestion.

In Section 2.3 Dataset specified in the “Training and Test” paragraph: we described how many slices were selected for training from control volumes. In manual segmentation we labeled three different classes (background, transverse and longitudinal fibers). A demonstrative figure was added to show it. The AI model’s performance would evaluate only with accuracy score and visualizing the application on all 3D volumes.

  1. Line 146: Is this threshold set for all tissue volumes, including the augmented data? Moreover, does this method set a threshold for each sample in the dataset individually, or evaluate samples together and then set the threshold? Please address these questions in the paragraph.

Section 2.3 and 2.4: Data augmentation was applied in the online version (Ref. [27] was added in Line 180), inside the Dragonfly’s tool, therefore the threshold is not set for augmented data. In Lines 188-189 we added that the Otsu algorithm was applied to each volume selected (formed by 300 slices) and it chose a single threshold for each of them.

  1. Line 151: Please consider reevaluating this paragraph, especially from lines 151-154.

Lines 190-193 modified, also other parts in paragraph are now better specified.

  1. Section 2.5: The structure, including the input size, number of layers, and depth of the layers of the U-Net, needs to be specified. I'd recommend adding a figure for the structure of the U-Net used in the article.

Section 2.5: Reference (Ref. [31]) and details were added; moreover, the table describing the U-Net structure was updated.

  1. Authors correctly indicated that segmentation aims to label each image pixel with a corresponding color; however, they never showed the datasets and the targets for each pixel they are trying to reach. This is a serious flaw in their work as the readers expect a demonstration of the input dataset and the targets for each pixel. Please consider addressing this issue and elaborating on the dataset as well as the classes for them.

Figure 1 added in paragraph 2.3 to show the targets from the manual segmentation.

  1. Line 275: why the connectivity density is significantly higher in Pt2_st_no-wt_l than in other samples?

The connectivity density is significantly higher in Pt2_st_no-wt_l than in all the other samples. This is most likely attributable to the immaturity of the connective tissue present in this biopsy, as shown by the morphological evidence introduced in the new Fig. S2 of the Supplementary Material. The abundant presence of micro vessels (Fig. S2a) and the presence of collagen fibers still strongly disorganized (Fig.S2b) at a three-dimensional level suggest the presence of a connective tissue only at the beginning of the remodeling phase, with a wound not yet completely healed at the connective level.

  1. Section 3.1: The fact that validation accuracy is higher than the training accuracy raises questions about the data selection for the input. Please elaborate on how the input data was randomized to avoid the model being biased. Also, normalizing the data could help the model perform better; maybe the authors could use normalized data as input and compare the results. Furthermore, why not use cross validation in evaluating the accuracy? This could help overcome the issue of having a small dataset. 

In addition, using augmented data is not creating an entirely new dataset, as the images are still related. Therefore, it is very important to elaborate on the data augmentation types used. Moreover, was a sample from a patient kept separately from the training and validation dataset to be used as a test for unseen data? Or all the images from the dataset were used in training and validation?

The fact that validation accuracy is higher than the training accuracy has been addressed in Section 3.1 and the data selection for the input has been better explained in Section 2.3. The randomization and the separation in training and validation sets are internally handled by the AI tool in Dragonfly.

The normalization operation was not taken into consideration because the starting dataset was extremely homogeneous due to the same experimental settings (listed in M&M). In follow-up experiments, when considering larger datasets, we will consider doing a normalization as suggested by the reviewer.

Data augmentation methods and dataset creation are now better described in Section 2.3.

  1. Section 3.2: Do the results, especially in figure 4, indicate the Semantic segmentation performs better than the Otsu thresholding? Please care to elaborate on this section or the discussion.

Semantic and Otsu thresholding segmentation were compared in order to verify if  the results obtained by AI with semantic segmentation method were superimposable with those obtained by common thresholding techniques.  Referring to the volume percentage and to the degree of anisotropy, in general, the average values obtained by deep learning through semantic segmentation were similar to those obtained by the thresholding segmentation method. Conversely, referring to the connectivity density, the average values obtained by deep learning through semantic segmentation were significantly higher than those obtained by the Otsu thresholding segmentation method in all the samples but the Ctr1_st_wt. This higher level of connectivity is probably due to a better discrimination of the single fibers: consequently, each intertwinement is better recognized. These concepts are now made clearer throughout the Section 3.2.

  1. Section 3.3: The purpose of the statistical analysis performed in this section is unclear to the reviewer. Are the transversal bundles and longitudinal bundles compared for each sub-volume? The results from figure 5 do not make too much sense. For example, do we have the background in volume percentage as well? If not, then the sum of the transversal and longitudinal bundles should be 100% which is not in this figure. 

Using statistical analysis, are we trying to compare different properties of the sub-volume based on the transversal and longitudinal bundles identified by the AI segmentation? 

The transversal bundles and longitudinal bundles were extracted for each sample in four subvolumes in connective tissue close to the interface with the implant and then mean ± std.dev were calculated for each parameter(Vol.%, DA, Conn.D); afterwards, statistical analysis was performed between samples.

Figures 5-6 were modified in the new manuscript.

The 100% volume is constituted by background + longitudinal fibers + transversal fibers. It is also important to know what percentage volume is occupied by the background, made up of cells (fibroblasts) vessels and other components, because these data give us important information on the maturity level of the tissue.

  1. Line 319: How did the polarized light microscopy confirm the semantic segmentation results? Maybe it would be better to add the results from the semantic segmentation alongside the microscopy images for comparison.

The polarized light microscopy gives only qualitative results, while synchrotron radiation method gives volumetric information with quantitative morphometric results. However, polarized light microscopy qualitatively confirmed  the semantic segmentation results in both transversal and longitudinal sections.

  1. Line 403: The authors have drawn an interesting conclusion about the Ctr1_st_wt sample. However, this is very brief, and I would suggest discussing and emphasizing this conclusion with more details in their manuscript.

We discussed more extensively this aspect in the revised manuscript; clearly, given the demonstrative nature of this study and the extremely small sample size, this new finding will be verified within a wider sample size in follow-up studies.

Reviewer 2 Report

Reviewer’s Comments:

The manuscript “Collagen bundles rearrangement during wound healing in peri-implant soft tissues: an innovative application of Artificial Intelligence to synchrotron radiation-based imaging” is a very interesting work. In this work, the study of the organizational kinetics in the area surrounding the transmucosal part of dental implants promises to ensure an accurate diagnosis of the healing process, in terms of osseointe-gration and long-term implant success. In this demonstrative work, the morphological, qualitative and quantitative characteristics of 3D images of collagen bundles obtained by histology and synchrotron-based high-resolution X-ray tomography were analysed. In particular, the synchrotron data analysis was performed using deep learning algorithms, neural networks that were applied on multiple volumes extracted from connective portions of different patients. These volumes were selected within the portion of tissue showing more advanced maturation, therefore better organization of the collagen bundles. The neural network was trained from mutually consistent examples from different patients; in particular, we used a neural network model, U-Net, well established in image application of deep learning.While I believe this topic is of great interest to our readers, I think it needs major revision before it is ready for publication. So, I recommend this manuscript for publication with major revisions.

1. In this manuscript, the authors did not explain the importance of the wound healing in the introduction part. The authors should explain the importance of wound healing.

2) Title: The title of the manuscript is not impressive. It should be modified or rewritten it.

3) Correct the following statement “Once trained, it distinguished the collagen fibres from the background, but also subdi-vided the collagen bundles based on the orientation of the fibres. In fact, the presence of collagen bundles organized in an intertwining pattern indicates an advanced stage of healing. In conclusion, Artificial Intelligence has been shown to accelerate and improve the process of identification of collagen fibres during wound healing by precisely quantifying morphometric and 3D complexity parameters with the aim of monitoring, optimizing and accelerating the healing process based on the reorganization of collagen bundles”.

4) Keywords: The keywords should be small. So, modify the keywords.

5) Introduction part is not impressive. The references cited are very old. So, Improve it with some latest literature.

6) The authors should explain the following statement with recent references, “in this way, it was proved that the network learned without incurring in overfitting”.

7) Please justify the following statement “In order to show possible difference between thresholding and semantic segmentation, both methods were applied, evaluated and compared”.

8) The author should provide reason about this statement “In general, referring to Pt2 (i.e. both Pt2_st_no-wt_r and Pt2_st_no-wt_l samples), high standard deviations were found, both considering the volume percentage and the connectivity density; this was not the case of the two Ctr1, Ctr2 and Pt1 samples”.

9) Comparison of the present results with other similar findings in the literature should be discussed in more detail. This is necessary in order to place this work together with other work in the field and to give more credibility to the present results.

10) Conclusion part is very long. Make it brief and improve by adding the results of your studies.

11) There are many grammatic mistakes. Improve the English grammar of the manuscript.

Author Response

Dear reviewer,

We thank you for the observations made to our study which allowed us to complete and improve it. We have answered all the questions point by point. We hope that the changes introduced will prove to be a valid answer to your questions.

1. In this manuscript, the authors did not explain the importance of the wound healing in the introduction part. The authors should explain the importance of wound healing. 

Dear Reviewer, we improved the Introduction Section expanding the biomedical background of the research and introducing the importance of wound healing, also focusing on the peri-implant context.

2. Title: The title of the manuscript is not impressive. It should be modified or rewritten it.

This is the new Title: “Deep learning for Microstructural Characterization of Synchrotron Radiation-based Collagen Bundles Imaging in Peri-Implant Soft Tissues”. We believe it could work better.

3. Correct the following statement “Once trained, it distinguished the collagen fibres from the background, but also subdi-vided the collagen bundles based on the orientation of the fibres. In fact, the presence of collagen bundles organized in an intertwining pattern indicates an advanced stage of healing. In conclusion, Artificial Intelligence has been shown to accelerate and improve the process of identification of collagen fibres during wound healing by precisely quantifying morphometric and 3D complexity parameters with the aim of monitoring, optimizing and accelerating the healing process based on the reorganization of collagen bundles”.

Following the indication of the other Reviewer we completely modified the Abstract: we hope that the new version better matches with the manuscript contents.

4. Keywords: The keywords should be small. So, modify the keywords.

We did a keyword-search in Applied Sciences journal: two of our keywords were changed; the others have been left unchanged because they were included and accepted in different manuscripts.

5. Introduction part is not impressive. The references cited are very old. So, Improve it with some latest literature.

We improved the Introduction Section expanding the biomedical background of the research, also introducing new recent references. We also explained the relevance of applying AI models to predict the wound healing process in the tissue around the implant.

6. The authors should explain the following statement with recent references, “in this way, it was proved that the network learned without incurring in overfitting”.

Citation added to explain this concept: Ref. [33].

7. Please justify the following statement “In order to show possible difference between thresholding and semantic segmentation, both methods were applied, evaluated and compared”.

Statement modified in Section 3.2: “Semantic and Otsu thresholding segmentation were compared in order to verify if the results obtained by AI with semantic segmentation method were superimposable with those obtained by common thresholding techniques.”

8. The author should provide reason about this statement “In general, referring to Pt2 (i.e. both Pt2_st_no-wt_r and Pt2_st_no-wt_l samples), high standard deviations were found, both considering the volume percentage and the connectivity density; this was not the case of the two Ctr1, Ctr2 and Pt1 samples”.

The biopsies extracted from patient 2, in particular the one referring to the left side of the jaw, have very particular characteristics with high volume percentages and connectivity in the longitudinal direction and low anisotropy degree. This morphometric evidence is compatible with a probable immaturity of the connective tissue, most likely linked to the patient's clinical history. This evidence is now highlighted in Fig. S2 of the Supplementary Material. The abundant presence of micro vessels (Fig. S2a) and the presence of collagen fibers still strongly disorganized (Fig. S2b) at a three-dimensional level suggest the presence of a connective tissue only at the beginning of the remodeling phase, with a wound not yet completely healed at the connective level. We added these concepts in the manuscript.

9. Comparison of the present results with other similar findings in the literature should be discussed in more detail. This is necessary in order to place this work together with other work in the field and to give more credibility to the present results.

Thank you for this question. This is the first study referable to peri-implant connective tissue studied using artificial intelligence algorithms. We specified it better in the introduction, emphasizing how the need to differentiate the transverse and longitudinal collagen bundles had emerged from our previous work (Ref. [9]). However, conventional image segmentation methods, i.e. those that do not make use of artificial intelligence, do not allow this to be done.

10. Conclusion part is very long. Make it brief and improve by adding the results of your studies.

We improved the conclusion by adding the results of our studies. Unfortunately, we could not make it brief because the other reviewer asked to add some specification.  

11. There are many grammatic mistakes. Improve the English grammar of the manuscript.

The English grammar of the manuscript has been improved.

Round 2

Reviewer 1 Report

Thank you for addressing my comments. The paper looks presentable now, but I recommend another round of editing for English before the final publication.

Author Response

Another round of editing for English was done with an expert native speaker.